# Racism, Chronic Disease, and Mental Health: Time to Change Our Racialized System of Second-Class Care

**DOI:** 10.3390/healthcare9101276

**Published:** 2021-09-27

**Authors:** Judith L. Albert, Claire M. Cohen, Thomas F. Brockmeyer, Ana M. Malinow

**Affiliations:** 1Independent Researcher, Pittsburgh, PA 15215, USA; 2Southwood Psychiatric Hospital, Pittsburgh, PA 15241, USA; claire_cohen@me.com; 3Independent Researcher, Pittsburgh, PA 15208, USA; thmasbrockmeyer53@gmail.com; 4Department of Pediatrics, University of California, San Francisco, CA 98158, USA; ana.malinow@ucsf.edu

**Keywords:** racism, weathering hypothesis, adverse childhood experiences, mental health disparities, Medicaid, national single payer, Medicare for All

## Abstract

In this article, we describe how the “weathering hypothesis” and Adverse Childhood Experiences set the stage for higher rates of chronic disease, mental health disorders and maternal mortality seen in African American adults. We illustrate the toll that untreated and overtreated mental health disorders have on Black individuals, who have similar rates of mental health disorders as their white counterparts but have fewer outpatient mental health services and higher rates of hospitalizations. We discuss the history of Medicaid, which, while passed alongside Medicare during the Civil Rights era, was Congress’s concession to Southern states unwilling to concede federal oversight and funds to the provision of equal healthcare for poor and Black people. Medicaid, which covers 33% of all Blacks in the US and suffers from chronic underfunding and state efforts to weaken it through demonstration waivers, is a second-class system of healthcare with eligibility criteria that vary by state and year. We propose the adoption of a national, single payer Medicare for All system to cover everyone equally, from conception to death. While this will not erase all structural racism, it will go a long way towards leveling the playing field and achieving greater equity in the US.

## 1. Introduction

On 25 May 2020, Derek Chauvin knelt on George Floyd’s neck and back for 9 minutes, 29 seconds, killing him during an arrest in Minneapolis for allegedly using a counterfeit $20 bill. Nearly 2000 miles to the west, a healthy full-term boy was born in San Francisco to Stephanie Mills (not her real name). Stephanie, a 29-year-old Black woman with hypertension, named her son George. At the 1-month well-baby check, Stephanie scored off-the-chart high on her post-partum depression screen, including thoughts of self-harm. The pandemic meant isolation from any support, including her own parents, each of whom suffered from pre-existing conditions, putting them at greater risk for COVID-19. The images of George Floyd added to her anxiety as a new mother. She cried and grieved for George Floyd. The obstetrician found it hard to stabilize Stephanie’s hypertension. However, the next available psychiatrist appointment to manage Stephanie’s suicidal ideation for a patient on Medicaid was months away. Taken together, the circumstances surrounding the death of the Black man and the birth of the Black child are linked by the difficulties experienced by Stephanie, the young mother. She struggled with a social burden that may impact the course of her son’s life and health. Both George Floyd and Stephanie, the new mom, likely suffer from the weathering effects of living Black in America, as we will describe within.

## 2. The Weathering Hypothesis and Adverse Childhood Events

The “weathering hypothesis” was first proposed by Arline Geronimus in 1992: “Namely, that the health of African American women may begin to deteriorate in early adulthood as a physical consequence of cumulative socioeconomic disadvantage.” [1] Now, nearly 30 years later, a large body of literature has expanded upon this hypothesis. There is accumulating evidence that structural as well as interpersonal racism contribute to the significant increases in Black maternal and infant morbidity and mortality compared to that of whites. [2] Furthermore, this evidence is not limited to maternal-child health. [3] It is useful to focus on hypertension, as both George Floyd and Stephanie Mills suffered from chronic hypertension. Indeed, Derek Chauvin’s defense centered on the assumption that George Floyd’s pre-existing health conditions, including hypertension, were the actual cause of his death, not the asphyxiation from the knee on his neck.

In the US, physicians have been taught for generations that African Americans were genetically pre-disposed to hypertension, but documentation of an inheritance pattern derived from sub-Saharan African populations has been difficult to identify. Some studies indicate a gradient of hypertensive risk from low in African populations to moderate in Afro-Caribbean populations to highest in African Americans. [4]. It is now widely accepted that hypertension is manifested differently in US populations based on social determinants of health (SDoH) [5], defined broadly as the conditions in the environments where people are born, live, learn, work, play, worship, and age that affect a wide range of health, functioning, and quality-of-life outcomes and risks [6].

George Floyd’s health issues are symptomatic of a life lived with the effects of weathering and SDoH, and the newborn George will have a path through childhood, adolescence and adulthood that likely will be negatively impacted by the same risks. In addition to the impact of SDoH, including structural racism, his life may be further complicated by adverse childhood experiences (ACEs), which have also been shown to exert a cumulative effect on childhood, adolescent and adult physical and mental health [7]. Adverse childhood experiences include living with a household member who was depressed, mentally ill, attempted suicide, died or was incarcerated, and being treated or judged unfairly because of one’s or race or ethnic group. Clearly, racism can be considered an adverse childhood experience [8]. Data from the 2017–2018 National Survey of Children’s Health Report reveal that by age 18, one in three children has experienced at least one parent-reported ACE in their lifetime, and one in five has experienced two or more [9]. Given the household makeup of many Black children in America, it is disturbing, but not surprising, that among children whose parents make between 200 and 400 percent of the federal poverty level (in other words, the median household income), almost 20% of Black children, compared to 12% of white children, have experienced two or more ACEs in their lifetime [9]. Extensive research shows that childhood adversity, including excessive or prolonged activation of the stress response, changes our biological systems [10], putting children with higher ACE scores at higher risk of adverse health outcomes [7]. As with SDoH, a high ACE score likely plays a role in the “weathering effect.”

## 3. Mental Health and Maternal Health Disparities

According to the American Psychiatric Association, rates of mental illness among adults aged 18 or older are no different in Blacks than those of the general population [11], yet disparities exist in mental health service use: Blacks use about half the outpatient care and fewer psychiatric medications as white adults, yet are hospitalized at twice the rate as their white counterparts for mental illness [12]. Furthermore, while the occurrence of severe mental illness among Blacks is lower than among whites, Blacks with the same symptoms as whites are more likely to be diagnosed with schizophrenia, a disorder with greater stigma and worse prognosis, and less likely to be diagnosed with mood disorders [13]. Fear of being mis- or over-diagnosed might be at the root of the “stigma” so often attributed as the reason Blacks do not seek or use mental health services. When studied, few racial/ethnic differences are found among the reasons for not using mental health services, with service cost or lack of insurance as the most frequent reason for not seeking care among all racial/ethnic groups, and a belief that services did not help as the least cited reason [12]. In Stephanie’s case, it was lack of access, and not “stigma,” that prevented her from getting the mental health care she needed.

Finally, maternal mortality rates in the US far surpass those of other high-income countries, yet pregnancy-related mortality rates among Black women are three times higher than white women, even among college-educated Black women. The rate of maternal mortality among Black women aged 30–34 widens to four times higher than white women [14], confirming the weathering hypothesis that the health of African American women begins to deteriorate in early adulthood.

The equitable implementation of interventions requires access to physical and mental health services as well as early diagnosis and intervention, including comprehensive health insurance coverage, and this is specifically where equity fails Black Americans.

## 4. Disparities in Health Insurance Coverage Result in Second-Class Care

While the most prevalent form of health insurance in the US is employer-sponsored health insurance, only 47% of Black families, compared to 66% of white families, obtain their insurance from their employers [15]. This leaves most non-elderly, non-military Blacks with two options: public insurance or none at all. In fact, Blacks have the highest percentage of Medicaid enrollment of any racial or ethnic group other than American Indian and Alaskan Natives [16]. This makes Medicaid, a means-tested program for poor children and adults, a significant form of health insurance for Black Americans.

From its inception in 1965, conservative members of Congress, many from Southern states, opposed the idea of federal control of health programs for poor and low-income people, but found state control of Medicaid more acceptable. Medicaid historically allowed individual states to control access to medical care through means testing (eligibility based on income, age, pregnancy and disability) and has been funded differently in Southern states with higher Black populations, effectively imbedding racism into its structure. Indeed, states have been able to resist efforts to mandate universal standards for healthcare provision for poor children and adults at key points for over 50 years [17]. In the 1970s, Congress successfully resisted a provision in the Medicaid legislature that required states to provide comprehensive care to all who met the requirements, and in 2012, the US Supreme Court decision made Medicaid expansion in all states optional instead of mandatory, a decision that has impacted Black populations significantly in the Southern states with large populations of color [17]. These are also among the 12 states that have failed to expand Medicaid [18]. As a result of state control, Medicaid remains poorly funded and reimbursed, and is shunned by many medical providers [19]. Of the 14 states that have not expanded or have not yet implemented the expansion of Medicaid, 12 are among the states with the worse overall health system performance [20]. There is no denying that access to some health insurance is better than no access at all. However, when compared to those with private insurance, those with Medicaid report greater likelihoods of not having a usual source of care, postponing care, going without care, or delaying filling a prescription due to cost [21].

The lack of access to mental health providers is a critical issue for Medicaid subscribers: mental health providers may choose not to accept Medicaid patients because to do so is an economic hardship. Compared to reimbursements for primary care providers, reimbursements to psychiatrists are 24% lower, and in 11 states, the reimbursement to psychiatrists is over 50% lower [22]. As a result, the percent of psychiatrists that accept private health insurance is lower than other specialties and has declined by 17 % since 2005–2006, but the situation is even more dire for publicly insured individuals, as only 43% of psychiatrists accept Medicaid [23]. This is the case despite advocacy from the American Psychiatric Association for full insurer compliance with the 2008 Mental Health Parity and Addiction Equity Act that requires insurers to reimburse mental health care at a level comparable to physical health.

Medicaid, documented to improve maternal morbidity and mortality [24], covers 40% of all births in the US plus maternal care up to 60 days post-partum. After this period, it is up to the state to determine what happens to a mother’s health care needs, and as a result, millions of women lose health insurance shortly after delivery. The rate of perinatal “churning,” or losing health insurance prior to conception or in the post-partum period, is over 30%, with higher rates (41%) in non-Medicaid expansion states, including Southern states with greater numbers of people of color [25]. In expansion states, post-partum women may be eligible to purchase private health insurance in the Exchanges, even with premium subsidies, but many find coverage unaffordable and will likely have to change their provider [26]. In non-Medicaid expansion states, such as Texas, that have limited eligibility for adults with children, a post-partum woman on Medicaid must make under $2961 per year to qualify for Medicaid 60 days after delivery [27].

Section 1115 of the Social Security Act gives authority to the Secretary of Health and Human Services to approve “experimental, pilot or demonstration projects likely to assist in promoting the objectives of the Medicaid program.” [28]. This permits the Secretary to waive certain federal Medicaid requirements, allowing states greater flexibility to design and improve their programs. Historically, states have used these demonstration waivers to create programs that expand care to new populations, offer more services and expand delivery of care. However, since the passage of the Affordable Care Act, many states have used these waivers to restrict and limit, rather than expand and ease, access.

The Henry J. Kaiser Family Foundation compiled a tracker [29] that aggregates information on pending and approved Section 1115 Medicaid waivers. As of 28 June 2021, 63 waivers, across 45 states, had been approved and 27 waivers, across 22 states, were awaiting approval.

While a few waivers, such as expanding access to mental health and substance use care, promote Medicaid program objectives, others, such as work requirements, a priority under the Trump administration, are either pending, in Court, or already implemented in 15 states [29]. Studies show that 59% of Medicaid enrollees work, and 80% live in a family where at least one family member works. For those adults who do not work, most report an inability to work due to illness or disability, having to care for someone in their home, attending school, looking for work, or being retired [30]. Recognizing that these work requirements do not foster Medicaid program objectives, but instead lead to disenrollment, the Biden Administration sent letters to all states with approved work requirement directing them to begin the process to withdraw such requirements [31].

Demonstration projects waiving required benefits, imposing copays above statutory limits, fees for missed appointments, restrictions of free choice of family planning provider, and healthy behavior provisions have been approved or are pending in 17 states [31]. An article published by the Kaiser Family Foundation showed premiums to be a barrier to accessing and receiving services for low-income individuals, and cost-sharing amounts as low as $1 to $5 result in reduced use of care, including necessary services, increased use of emergency room care, and negative health consequences such as uncontrolled hypertension, elevated cholesterol and decreased asthma care for children [32]. They also found that any potential revenue gains to the state from premiums and copays collected from Medicaid enrollees are offset by disenrollment, increased use of emergency room care, greater costs for the uninsured, and added administrative expenses. Responding to Medicaid restrictions, six medical organizations, including the American Academy of Family Physicians, the American Academy of Pediatrics, the American College of Obstetrics and Gynecology, the American College of Physicians, the American Osteopathic Association, and the American Psychiatric Association, representing over 560,000 physicians and medical students, published a statement of joint principles cautioning state and federal authorities on the misuse of Medicaid waivers, exhorting them to “first, do no harm.” [33]. It is clear that leaving decisions as important as healthcare to states has resulted in mixed results, at best.

Perhaps most significantly and insidiously, the creation of Medicaid, which, as we have seen, functions as a second-class system for poor and low-income populations, has taught at least three generations of physicians, medical students and healthcare workers that it is acceptable to offer less treatment (fewer specialists within a network) at lower reimbursement rates than the more fortunate people with employer-based health insurance. From the very beginning of their medical education, trainees are “taught” that a system outside of the main private insurance healthcare is good enough for poor and low-income populations that are over-represented by Black populations. Unable to find providers that accept their public health insurance, Medicaid enrollees are shunted to the academic centers, which have traditionally accepted these patients in their “teaching” hospitals and clinics. In our experience, it is not unusual for a patient at an academic institution to spend 3 hours seeing a doctor in clinic, seeing first the medical student, then the resident and finally the attending physician, as if that patient’s time is not as valuable as that of the patient with private health insurance. Further, it can be argued that an entire system based on employer-sponsored health insurance is racist by design: to be in the position to be offered affordable health insurance by your employer requires a well-paid job, possibly a union job, a good education starting in pre-school, freedom from police harassment, and affordable and safe housing, all made difficult, if not impossible, by generations of laws that have advantaged the dominant class. By the time individuals on Medicaid reach the age of Medicare eligibility, they have already been subjected to a racist system that is structurally rigged against them. Furthermore, while health outcomes do improve for Blacks after age 65 and disparities between white and Black health outcomes narrow [34], a lifetime of racist experiences, and discrimination by medical professionals, in addition to the damaging effects of SDoH, have inflicted their damage.

The success of programs that could identify and address the chronic physical and mental health implications of SDoH, racism and the disparate distribution of ACEs between Black and white children requires a healthcare system that people of all races and incomes can access from birth to death. The employer-based health insurance system in the US is simply unable to do this, since many low wage workers are ineligible because they are not full-time employees, or they cannot afford the copays and deductibles in the plan offered by the employer. This leaves 30 million individuals without health insurance and another 40 million underinsured [35]. Medicaid offers coverage to the uninsured who qualify for benefits. However, as we have seen, this qualification, or means testing, creates significant barriers to access that vary state by state and year by year.

## 5. A More Equitable System under Medicare

In contrast, Medicare, a system also created in the midst of the civil rights movement, is universal based on age alone, and administered federally, not state by state: everyone is eligible regardless of race, economic status or employment at age 65. Nearly all physicians participate in Medicare and receive comparable levels of reimbursement for medical care.

No other high-income country has tiered their health care financing and delivery system as the US has: based not on need, but the ability to pay. Other countries with similar levels of economic development to the US have managed, some for almost a century, to create equitable universal access to health care with better outcomes and at less cost than we have achieved in the US.

If health care is a human right, as many Americans believe, given the strong public support for Medicare for All [36], it cannot be left up to states to determine who deserves this right and who does not. It is the federal government’s duty to guarantee this.

National single payer healthcare, popularly known as Medicare for All, is a publicly funded, privately delivered healthcare system, where government-run insurance replaces private insurance, Medicare, Medicaid (including Medicare Advantage and Medicaid Managed Care), and the Children’s Health Insurance Program, while TriCare, the Veteran’s Administration and the Indian Health Service remain intact. Taxes will inevitably be used to fund the program, but these will be offset by the elimination of premiums, copays, deductibles and coinsurance. Once instituted, doctors and patients would see very little disruption as there will be free choice of provider and hospital. There are currently two bills [37,38] proposed in Congress that detail the use of savings generated from lower administration costs and negotiated prices with pharmaceutical companies to establish a national health insurance program that covers all US residents for all medically necessary services, including prescription drugs and long-term care, and prohibits private health insurance companies from duplicating services offered by the national insurance program. As stated above, there is widespread popular support for expanding Medicare to all Americans, as well as institutional support from organizations such as the American College of Physicians [39], Physicians for a National Health Program [40], and National Nurses United [41]. In a systematic review of the economic analyses of 22 single payer plans, Cai and colleagues found consensus for the “fiscal feasibility” of a single-payer financing approach in the US [42].

## 6. Conclusions

How could a national single payer healthcare system in the US improve the health outcomes of baby George and his mother, Stephanie? Especially when faced with a world-wide pandemic? While a more equitable healthcare system will not dismantle all the structural racism that they will face, access to comprehensive, universal, high-quality healthcare will go a long way to level the playing field. The second-class, means-tested healthcare system, Medicaid, would no longer be necessary. A well-financed, government-run (and privately delivered) healthcare system would not be based on a profit-driven private insurance market. It would not be linked to employment. It would be accessible to all and accountable to the people it serves. It would institute just and fair reimbursement of providers from all specialties, including mental health and the long-term care of chronic disease. The effects of weathering and disparate distribution of ACEs will not disappear quickly, but recognition of, and access to, care for the resulting medical conditions would become widely available. A national single payer healthcare system offers the promise of the continuity of care over a lifetime. This is what baby George should look forward to, and what this country must achieve.

## Data Availability

Data supporting this commentary can be found in the References.

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
