# Peer review of "Racism, Chronic Disease, and Mental Health: Time to Change Our Racialized System of Second-Class Care"

_healthcare, 2021, doi:10.3390/healthcare9101276_

Round 1

Reviewer 1 Report

The social issue of racial justice relative to healthcare access is a very important topic and should be of interest to a wide range of readers.

The article is clearly written, though there are a few stylistic issues with sentence structure. For example, the opening line of the article is a bit confusing based on the names used and the syntax.  Perhaps break that into two sentences, or reorganize the paragraph to make clear how those separate events, of life and death, are connected by a shared suffering among black Americans.

Overall, I found the article compelling as a scholar of American religions. I am convinced by the call for universal health care access, expressed here as a single payer Medicare for all system. I am not a healthcare professional, however, so I cannot speak to the particularities of that plan. 

Related to my own research area, I am interested in the role religion plays in resisting human rights claims to universal health care. The article gestures in that direction, by describing Medicaid as a concession to southern states. Those states, which can be easily demonstrated through Pew data, have the highest rates of religious affiliation in the country, with the majority of white evangelical Protestants in the South and sunbelt. If possible, it could be helpful for the authors to add a paragraph that considers the role of religion in the passage of Medicaid. Did white evangelicals support it? Southern religion as white evangelicalism was a staunch supporter of racial segregation prior to the Civil Rights Act, and afterward turned its attention to conservative moral issues like abortion as a way to undercut, without saying it, racial justice legislation. Did Medicaid also allow states to bypass any federal regulation of women's healthcare access as well? If so, did supporters of Medicaid use any kind of claims to those family values moral issues, emphasizing the funding of abortion, as a way to hide the racial disparities outlined in the article?

These questions may be beyond the scope of the article. So I don't think revisions along these lines should be required for publication. However, if the authors can address some of them in a paragraph or two, it might provide more context to be understand why the U.S. doesn't have a single payer system, or anything close to universal health care.

Again, I really enjoyed reading this article and fully support its publication. 

Author Response

Thank you for your review of our submission "Racism, Chronic Disease, and Mental Health:  Time to Change our Racialized System of Second Class Care".  We appreciate your suggestions about sentence structure and syntax in the first paragraph.  We have re-written that paragraph by splitting the first sentence into two to clarify the birth and death events of the two individuals described.  We expanded the conclusion to connect the shared suffering as you suggested. 

Your comments about the role religion might have played in resisting human rights claims to universal healthcare are interesting and provocative.  We are not religious scholars, but we are aware of the many ways that religious organizations supported segregation and racist policies since the founding of the United States.  The influence of evangelical groups who support restrictions to access to abortion had significant impact on the creation of the ACA.  We agree that an analysis of the impact of religion is an important topic, one that is beyond the scope of our paper.  Answers to the important questions you posed would require in depth research and documentation in order to draw meaningful conclusions that go beyond our opinion.  We would love to see this type of analysis in another paper!  

Again, thank you for your insightful review.

Reviewer 2 Report

The manuscript is offered in a form of the commentary. It deals with the weathering hypothesis and it tries to assess the possibilities of the environment, where the hypotheses could be verified. The case of George Floyd that is taken into consideration looks like an opportunity to apply the weathering hypothesis in order to reveal whether his surrounding environment is able to multiply the health deterioration of an individual. The whole commentary is discussed in a very formal way. This would be beneficial, but however, some other references leading to the scientific sources would enhance the expressions and the thought presented in the manuscript. For instance, mostly, the cost-sharing evaluation according to the mentioned study by Artiga et al. Moreover, it is not included in the list of the references. Also, the references are marked by the upper index sign, though they should be placed in the square brackets. The end of the commentary should involve the concluding remarks. From a formal point of view, the text should be revised in order to avoid the double spaces for instance – on page 5, lines 7 and 12, the third occurrence in the third paragraph on line 3, the fourth occurrence in the sixth paragraph on line 8, and so on.

Author Response

Thank you for your review of our submission "Racism, Chronic Disease, and Mental Health: Time to Change our Racialized System of Second Class Care".  We have made changes to the statement referring to Artiga's work, which is actually contained in our reference number 32.  The reference is a publication by the Kaiser Family Foundation that does not explicitly contain her name, which is a source of confusion. The text has been changed to read:  "An article published by the Kaiser Family Foundation showed premiums to be a barrier to accessing and receiving services for low-income individuals, and cost-sharing amounts as low as $1 to $5 result in reduced use of care, including necessary services, increased use of emergency room care, and negative health consequences such as uncontrolled hypertension, elevated cholesterol and decreased asthma care for children.32" We agree that additional scientific sources would further support our conclusions, however, there are relatively few published studies that focus specifically on racism in medicine.  In fact, racism was only added as a searchable term in PubMed in 2013, indicating the historical lack of attention to this issue. 

I have attended to the double spaces.  I believe the format used for references is accepted by this publication. 

Again, thank you very much for your review and comments!